# The evaluation of COVID-19 prediction precision with a Lyapunov-like exponent

Jiří Mazurek *

Silesian University in Opava, School of Business Administration in Karvina, Karvina, Czech Republic

* mazurek@opf.slu.cz

## Abstract

In the field of machine learning, building models and measuring their performance are two equally important tasks. Currently, measures of precision of regression models' predictions are usually based on the notion of mean error, where by error we mean a deviation of a prediction from an observation. However, these mean based measures of models' performance have two drawbacks. Firstly, they ignore the length of the prediction, which is crucial when dealing with chaotic systems, where a small deviation at the beginning grows exponentially with time. Secondly, these measures are not suitable in situations where a prediction is made for a specific point in time (e.g. a date), since they average all errors from the start of the prediction to its end. Therefore, the aim of this paper is to propose a new measure of models' prediction precision, a *divergence exponent*, based on the notion of the *Lyapunov exponent* which overcomes the aforementioned drawbacks. The proposed approach enables the measuring and comparison of models' prediction precision for time series with unequal length and a given target date in the framework of chaotic phenomena. Application of the divergence exponent to the evaluation of models' accuracy is demonstrated by two examples and then a set of selected predictions of COVID-19 spread from other studies is evaluated to show its potential.

## Introduction

Making (successful) predictions certainly belongs among the earliest intellectual feats of modern humans. They had to predict the amount and movement of wild animals, places where to gather fruits, herbs, or fresh water, and so on. Later, predictions of the flooding of the Nile or solar eclipses were performed by early scientists of ancient civilizations, such as Egypt or Greece. The latter civilization gave birth to determinism, a philosophical view that all events in the future could be fully determined if we had knowledge of the current state of all matter and of all laws governing that matter [1].

However, at the end of the $19^{th}$ century, the French mathematicians Henri Poincare and Jacques Hadamard discovered the first chaotic systems and that they are highly sensitive to initial conditions. Small differences in initial conditions (due to errors in measurements or rounding errors) in such systems lead to widely diverging outcomes, rendering (precise) long-term predictions impossible in general [2]. Chaotic behavior can be observed in fluid flow,

**Data Availability Statement:** All relevant data are within the paper and its Supporting information files.

**Funding:** This paper was supported by the Ministry of Education, Youth and Sports Czech Republic

within the Institutional Support for Long-term Development of a Research Organization in 2021.

**Competing interests:** The authors have declared that no competing interests exist.

weather and climate, road and Internet traffic, stock markets, population dynamics, or a pandemic.

Since absolutely precise predictions (of not-only chaotic systems) are practically impossible, a prediction is always burdened by an error. The smaller this error, the more valuable and helpful the prediction, while bad predictions are not only useless, but can be even harmful [3].

The precision of a regression model prediction is usually evaluated in terms of explained variance (EV), coefficient of determination ($R^2$), mean squared error (MSE), root mean squared error (RMSE), magnitude of relative error (MRE), mean magnitude of relative error (MMRE), and the mean absolute percentage error (MAPE), etc., see e.g. [4, 5]. These measures are well established both in the literature and research, however, they also have their limitations. The first limitation emerges in situations when a prediction of a future development has a date of interest (a *target date*, *target time*). In this case, the aforementioned mean measures of prediction precision take into account not only observed and predicted values of a given variable on the target date, but also all observed and predicted values of that variable before the target date, which are irrelevant in this context. The second limitation, even more important, is connected to the nature of chaotic systems. The longer the time scale on which such a system is observed, the larger the deviations of two initially infinitesimally close trajectories of this system. However, standard (mean) measures of prediction precision ignore this feature and treat short-term and long-term predictions equally.

Therefore, the aim of this paper is to propose an alternative approach to the evaluation of prediction precision dealing with chaotic systems, where a prediction is related to a given target date, which utilizes the notion of the *Lyapunov exponent*, see [6, 7]. In analogy to the Lyapunov exponent, a newly proposed *divergence exponent* expresses how much a (numerical) prediction diverges from observed values of a given variable at a given target time, taking into account only the length of the prediction and predicted and observed values at the target time. The larger the divergence exponent, the larger the difference between the prediction and observation (prediction error), and vice versa. Thus, the presented approach avoids the shortcomings mentioned in the previous paragraph.

This new approach is demonstrated in the framework of the COVID-19 pandemic. After its outbreak, many researchers have tried to forecast the future trajectory of the epidemic in terms of the number of infected, hospitalized, recovered, or dead. For the task, various types of prediction models have been used, such as compartmental models including SIR, SEIR, SEIRD and other modifications, see e.g. [8–12], artificial neural network models [13–16], Gompertz and logistics functions [17–19], ARIMA models [13, 20], and many other approaches, see e. g. [21–24]. A survey on how deep learning and machine learning is used for COVID-19 forecasts can be found e.g. in [25, 26]. General discussion on the state-of-the-art and open challenges in machine learning can be found e.g. in [27].

Since a pandemic spread is, to a large extent, a chaotic phenomenon, and there are many forecasts published in the literature that can be evaluated and compared, the evaluation of the COVID-19 spread predictions with the divergence exponent is demonstrated in the numerical part of the paper.

The data sources for this study included the Worldometers website [28], University of John Hopkins resource center [29] and CDC (Centers for Disease Control and Prevention) database [30].

The paper is organized as follows: in Section 2 Lyapunov and divergence exponents are introduced and their application is demonstrated with examples, Section 3 provides a numerical evaluation of selected models' predictions, Section 4 is devoted to a discussion and the Conclusions section closes the article.

## Lyapunov and divergence exponents

The Lyapunov exponent quantitatively characterizes the rate of separation of (formerly) infinitesimally close trajectories in dynamical systems. Formally, the Lyapunov exponent is defined as follows [6, 7]:

**Definition 1**

*Let $\delta Z(t)$ be a separation vector of two trajectories in a given phase space at the time t and let $\delta Z(0)$ be an initial separation vector of the two trajectories at the time t = 0. Then, the Lyapunov exponent $\lambda$ is defined via the following equation*:

$$\delta Z(t) = e^{\lambda t} \cdot \delta Z(0). \tag{1}$$

Since physical systems are usually multi-dimensional, Lyapunov exponents from each dimension of a phase space form a spectrum, and the predictability of a system is determined by the Maximal Lyapunov exponent (MLE). The MLE is defined as follows [6, 7]:

$$\lambda_{MLE} = \lim_{t \to +\infty} \lim_{\delta Z_0 \to 0} \frac{1}{t} \ln \left| \frac{\delta Z(t)}{\delta Z_0} \right| \tag{2}$$

The higher the Lyapunov exponent, the more chaotic the given dynamical system. Lyapunov exponents for classic physical systems are provided e.g. in [6, 7, 31, 32].

A prediction of a pandemic spread and the real data about the spread can be analogically considered using two trajectories in a one-dimensional phase space (a $\Re_+$ space) which start at the time $t = 0$, when both trajectories are identical, and then they inevitably diverge at some time $t > 0$.

Drawing upon the analogy with the Lyapunov exponent in (1–2), we introduce a "divergence exponent" $\lambda$.

**Definition 2**

*Let $P(t)$ be a prediction of a pandemic spread (given as the number of infections, deaths, hospitalized, etc.) in the time $t > 0$, and let $N(t)$ be a true (observed) value of a pandemic spread in the time $t > 0$. Then, the divergence exponent $\lambda$ is given as*

$$P(t) = e^{|\lambda|t} \cdot N(t), \tag{3}$$

*which, after rearrangement, gives*

$$\lambda = \left| \frac{\ln \left( \frac{P(t)}{N(t)} \right)}{t} \right|. \tag{4}$$

The larger the $\lambda$, the worse the prediction. In the case of an absolutely precise prediction, $\lambda = 0$.

For the sake of comparisons with the $\lambda$, one of the most common measures of prediction precision is the *mean relative error* (*MRE*):

$$MRE = \frac{1}{n} \sum_{i=1}^{i=n} \left| \frac{P(i) - N(i)}{N(i)} \right| \tag{5}$$

where $P(i)$ denotes the predicted value and $N(i)$ the observed value at the point $i$.

The following (extremely oversimplified) example shows how the $\lambda$ is calculated and one of its virtues.

**Table 1. Fictional pandemic spread.** The variable $N(t)$ denotes observed new daily cases, $P(t)$ denotes the prediction of new daily cases, and $t$ is the number of days.

|          | t = 0 | t = 1 | t = 2 | t = 3 | t = 4 | t = 5 |
|----------|-------|-------|-------|-------|-------|-------|
| $N(t)$   | 1     | 1     | 1     | 1     | 1     | 1     |
| $P_1(t)$ | 1     | 2     | 4     | 8     | 16    | 32    |
| $P_2(t)$ | 1     | 1/2   | 1/4   | 1/8   | 1/16  | 1/32  |

Consider the pandemic spread from Table 1. At the beginning ($t = 0$) the variable $N(t)$, which denotes the observed number of new daily infection cases, is set to 1 unit (for example 1,000 people). Two prediction models, $P_1$, $P_2$ were constructed to predict future values of $N(t)$, for five days ahead. While $P_1$ predicts exponential growth by the factor of 2, $P_2$ predicts that the spread will exponentially decrease by the factor of 2. After the predictions are made, reality shows that the spread is constant in time for $t \in \{1, 2, 3, 4, 5\}$.

Now, let's evaluate the precision of the model $P_1$: $\lambda = \left| \frac{ln(\frac{P(t)}{N(t)})}{t} \right| = \left| \frac{ln(\frac{32}{1})}{5} \right| t = 0.693$. Values of the prediction $P_1(t)$ grow exponentially by the factor $\delta = 2$. The factor $\delta$ can be easily obtained from the $\lambda$, see relation (3), as follows:

$$\delta = e^{|\lambda|} = 2 \tag{6}$$

Therefore, from the divergence exponent $\lambda$ the coefficient $\delta$ can be reconstructed, and vice versa. The coefficient $\delta$ is a base of a corresponding power series expressing the divergence of a prediction.

Now, consider the prediction $P_2(t)$. This prediction is arguably equally imprecise as the prediction $P(t)$, as it provides values halving with time, while $P(t)$ provided doubles. As can be checked by formula (4), the divergence exponent for $P_2(t)$ is 0.693 again. Therefore, over-estimating and under-estimating predictions are treated equally.

However, when we calculate the MRE of both predictions, we obtain: $MRE(P_1) = 6.5$, while $MRE(P_2) = 0.766$, which suggests that the prediction $P_2$ is much better.

Another virtue of the evaluation of prediction precision with a divergence exponent is that it enables a comparison of predictions with different time frames, which is demonstrated in the following example.

Consider a fictional pandemic spread from Table 2. Again, in the time $t = 0$, $N(0) = 1$, the prediction model $P_3$ is built and gives predictions for $t$ from 1 to 8 days. We evaluate $\lambda$ and $MRE$:

$$\lambda(P_3) = \left| \frac{\ln\left(\frac{P(t)}{N(t)}\right)}{t} \right| = \left| \frac{\ln\left(\frac{256}{1}\right)}{8} \right| = 0.693.$$

$$MRE(P_3) = \frac{1}{8}\left( \frac{2-1}{1} + \frac{4-1}{1} + ... \right) = 62.75.$$

**Table 2. Fictional pandemic spread.** The variable $N(t)$ denotes observed new daily cases, $P(t)$ denotes the prediction of new daily cases, and $t$ is the number of days.

|          | t = 0 | t = 1 | t = 2 | t = 3 | t = 4 | t = 5 | t = 6 | t = 7 | t = 8 |
|----------|-------|-------|-------|-------|-------|-------|-------|-------|-------|
| $N(t)$   | 1     | 1     | 1     | 1     | 1     | 1     | 1     | 1     | 1     |
| $P_3(t)$ | 1     | 2     | 4     | 8     | 16    | 32    | 64    | 128   | 256   |

According to the MRE value, the prediction model $P_3$ is much worse than $P_1$ (see Example 1), since its MRE value is much higher. However, an attentive reader may have noticed that the model $P_3$ is exactly the same as the model $P_1$, but provides a prediction for three additional days (on the contrary, the divergence exponent $\lambda$ provides the same value for $P_1$ and $P_3$).

The root of the problem with different values of MRE for the predictions $P_1$ and $P_3$, which are in fact identical, rests in the fact that MRE does not take into account the length of a prediction, and treats all predicted values equally (in the form of the sum in (5)). However, the length of a prediction is crucial in forecasting real chaotic phenomena, since prediction and observation naturally diverge more and more with time, and the slightest change in the initial conditions might lead to an enormous change in the future (Butterfly effect). A weather forecast one hour ahead is easy, a forecast for three days ahead is difficult, a forecast for a week ahead is extremely difficult, and a forecast for one month ahead is impossible due to the chaotic behavior of the Earth's atmosphere. Therefore, since MRE and similar measures of prediction accuracy do not take into account the length of a prediction, they are not suitable for the evaluation of chaotic systems, including a pandemic spread.

## A comparison of selected COVID-19 predictions

In this section, selected predictions about the COVID-19 pandemic are evaluated and compared via the divergence exponent $\lambda$ (and the relative error RE) introduced in the previous section. There have been hundreds of predictions of the COVID-19 spread published in the literature so far, hence for the evaluation and comparison of predictions only one variable was selected, namely the total number of infected people (or *total cases*, abbr. *TC*), and selected models with corresponding studies are listed in Table 3. The selection of these studies was based on two merits: first, only real predictions into the future with the clearly stated dates $D_0$ and $D(t)$ (see below) were included, and, secondly, the diversity of prediction models was preferred.

The data in Table 3 include the model's number, name of the lead author, model's specification, forecasted country, date when the prediction was made ($D_0$), target date of the prediction ($D_t$), length of the prediction in days ($t$), predicted number of total cases at a target day ($P(t)$), observed number of total cases at a target day ($N(t)$), divergence exponent ($\lambda$) and relative error ($RE$).

Fig 1 provides a graphical comparison of results in the form of a scatterplot, where each model is identified by its number, and models are grouped into five categories (distinguished by different colors): artificial neural network models, Gompertz models, compartmental models, Verhulst models and other models. Two models' outputs (models 13 and 24) were identified as outliers, and were removed from Fig 1.

As can be seen both from Table 3 and Fig 1, the most successful prediction with respect to $\lambda$ was provided by models (8) and (28), while the worst prediction came from (24). The most successful model with respect to $RE$ was model (8) followed by model (2), while the worst predictions came from models (13) and (24).

Pearson's correlation coefficient between $\lambda$ and $RE$ was 0.55, indicating a medium strength of the linear relationship between both variables.

## Discussion and limitations of the proposed approach

The evaluation of models' prediction precision by the divergence exponent (4) was illustrated in Section 3. Twenty-eight models' predictions with target dates published recently in the literature were evaluated and compared, see Table 3 and Fig 1. The primary purpose of this evaluation was to show the application and potential of the divergence exponent, not to draw some

**Table 3. The evaluation of prediction precision for selected models.**

| Study | Country | Model | $D_0$ | $D(t)$ | $t$ | N(t) | P(t) | λ | RE |
|---|---|---|---|---|---|---|---|---|---|
| (1) Mazurek [18] | World | Gompertz | May 2 | July 13 | 72 | 13,631,634 | 8,000,000 | 0.0074 | 0.413 |
| (2) Mazurek [18] | UK | Gompertz | May 2 | June 30 | 59 | 283,253 | 278,100 | 0.0003 | 0.018 |
| (3) Mazurek [18] | Russia | Gompertz | May 2 | June 3 | 34 | 432,277 | 300,000 | 0.0107 | 0.306 |
| (4) Li, M. [21] | Italy | PSO | March 13 | March 31 | 18 | 105,776 | 100,000 | 0.0031 | 0.055 |
| (5) Li, M. [21] | Iran | PSO | March 13 | March 31 | 18 | 44,605 | 33,000 | 0.0167 | 0.260 |
| (6) Li, L. [22] | World | Eureqa | March 1 | May 22 | 82 | 5,522,504 | 5,746,000 | 0.0005 | 0.040 |
| (7) Koczkodaj [33] | World | LS | March 18 | March 31 | 13 | 878,998 | 1,000,000 | 0.0099 | 0.138 |
| (8) Sanchez-Cab. [23] | Spain | Verhulst | April 4 | April 30 | 26 | 231,531 | 230,000 | 0.0003 | 0.007 |
| (9) Sanchez-Cab. [23] | Italy | Verhulst | April 4 | April 30 | 26 | 205,449 | 200,000 | 0.0010 | 0.027 |
| (10) Sanchez-Cab. [23] | France | Verhulst | April 4 | April 30 | 26 | 129,581 | 189,000 | 0.0145 | 0.459 |
| (11)Sanchez-Cab. [23] | UK | Verhulst | April 4 | May 30 | 56 | 247,171 | 160,000 | 0.0078 | 0.353 |
| (12) Bedi [9] | USA | SEIRD | Sep. 9 | Dec. 31 | 113 | 20,306,221 | 12,337,873 | 0.0044 | 0.392 |
| (13) Bedi [9] | India | SEIRD | Sep. 9 | Dec. 31 | 113 | 10,267,283 | 33,994,849 | 0.0106 | 2.311 |
| (14) Bedi [9] | Brazil | SEIRD | Sep. 9 | Dec. 31 | 113 | 7,619,970 | 6,947,629 | 0.0008 | 0.088 |
| (15) Bedi [9] | Russia | SEIRD | Sep. 9 | Dec. 31 | 113 | 3,159,297 | 1,576,715 | 0.0062 | 0.501 |
| (16) Arias [17] | USA | Gompertz | April 5 | April 25 | 20 | 980,753 | 1,000,000 | 0.0010 | 0.020 |
| (17) Sun [12] | USA | D-SEIQ | April 27 | May 27 | 30 | 1,783,730 | 1,375,000 | 0.0087 | 0.229 |
| (18) Sun [12] | Italy | D-SEIQ | April 27 | May 27 | 30 | 231,138 | 207,000 | 0.0037 | 0.104 |
| (19) Sun [12] | France | D-SEIQ | April 27 | May 27 | 30 | 145,746 | 156,000 | 0.0023 | 0.070 |
| (20) Sun [12] | Germany | D-SEIQ | April 27 | May 27 | 30 | 181,895 | 177,000 | 0.0009 | 0.027 |
| (21) Gupta [11] | India | SEIR | May 10 | May 31 | 21 | 190,648 | 174,293 | 0.0043 | 0.086 |
| (22) Gupta [11] | India | Regression | May 10 | May 31 | 21 | 190,648 | 205,768 | 0.0036 | 0.079 |
| (23) Tamang [14] | USA | ANN | May 9 | May 18 | 10 | 1,464,232 | 1,955,865 | 0.0289 | 0.336 |
| (24) Tamang [14] | France | ANN | May 9 | May 18 | 10 | 140,036 | 342,272 | 0.0894 | 1.444 |
| (25) Devaraj [13] | World | ARIMA | May 9 | June 30 | 52 | 10,417,063 | 9,493,908 | 0.0018 | 0.087 |
| (26) Devaraj [13] | World | LSTM | May 9 | June 30 | 52 | 10417063 | 9400000 | 0.0020 | 0.098 |
| (27) Devaraj [13] | World | SLSTM | May 9 | June 30 | 52 | 10,417,063 | 9,900,000 | 0.0010 | 0.050 |
| (28) Devaraj [13] | India | SLSTM | May 9 | June 30 | 52 | 3,679,782 | 3,800,000 | 0.0003 | 0.033 |

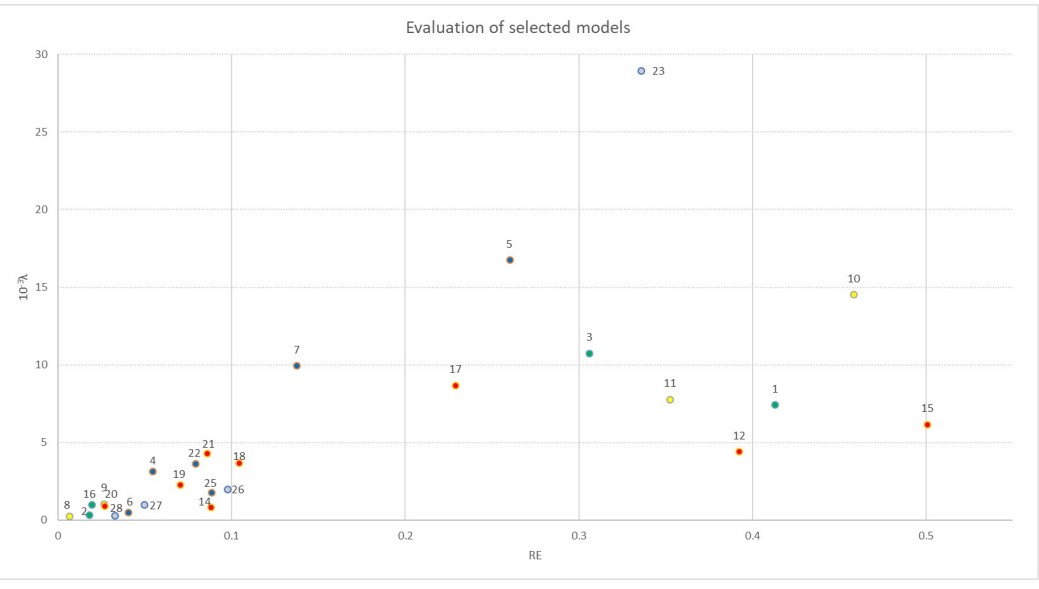

**Fig 1.**

general conclusions about models' performances in predicting the COVID-19 spread. This would require significantly more data. Since the pandemic is not over, there are undoubtedly many forecasting studies yet to be published, hence a comprehensive study on models' performance with respect to the COVID-19 spread is conceivable in the near future.

Though the evaluation of models' prediction precision with the divergence exponent can be applied in many other scientific fields where chaotic phenomena emerge, it has its limitations. It should be used only under specific circumstances, namely when a (numerical) characteristic of a chaotic system is predicted over a given time-scale and a prediction at a target time is all that matters. There are many situations where these circumstances are not satisfied, hence the use of the divergent exponent would not be appropriate. Consider, for example, daily car sales to be predicted by a car dealer for the next month. Suppose that the car dealer sells from zero to three cars per day, with two cars being the average daily sale. In this case, all days of the next month matter, and it is unrealistic to assume that sales at the end of the next month may reach hundreds or thousands, thus diverging substantially from the average.

In addition, standard measures of prediction precision (or rather prediction error), such as MAPE, have a nice interpretation in the form of a ratio, or a percentage. If, for example, MAPE = 8%, it means that a prediction deviates from an observation by 8% and an expert can conclude that the prediction was successful, since, according to the rule of thumb by Lewis [34], a prediction with MAPE under 10% is considered highly accurate. On the other hand, a prediction with MAPE over 50% is considered inaccurate [34]. Currently, there is no similar rule of thumb for the divergence exponent, so knowing its value does not provide a modeller with explicit information about the model's performance. Information acquired from the divergence exponent provides, however, a way for a *relative* comparison of different models' performances.

## Conclusions

In this paper, a new measure of prediction precision for regression models and time series, a *divergence exponent*, was introduced. This new measure has two main advantages. Firstly, it takes into account the time-length of a prediction, since the time-scale of a prediction is crucial in the so-called chaotic systems. Secondly, it evaluates the model's prediction performance only with respect to the end time of the prediction (a target time, or a target date), and the final deviation of the prediction from the observation.

Models' performance evaluation with the divergence exponent was illustrated on predictions of the COVID-19 spread published recently in the literature. Altogether, twenty-eight different models were compared. Verhulst and Gompertz models performed among the best, but no clear pattern revealing the types of models that performed best or worst was found.

The future research can focus on a comparison of different kinds of machine learning models in different environments where chaotic systems prevail, including various fields, such as epidemiology, engineering, medicine, or physics.

## Supporting information

**S1 Data.**
(XLSX)

## Author Contributions

**Conceptualization:** Jiří Mazurek.

**Data curation:** Jiří Mazurek.

**Formal analysis:** Jiří Mazurek.

**Investigation:** Jiří Mazurek.

**Methodology:** Jiří Mazurek.

**Validation:** Jiří Mazurek.

**Visualization:** Jiří Mazurek.

**Writing – original draft:** Jiří Mazurek.

**Writing – review & editing:** Jiří Mazurek.

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
