## [Decision Letter · Decision Letter 0]

12 Apr 2021

PONE-D-21-09821

The Evaluation of COVID-19 Predictions' Precision with a Lyapunov-like exponent

PLOS ONE

Dear Dr. Mazurek,

Thank you for submitting your manuscript to PLOS ONE. After careful consideration, we feel that it has merit but does not fully meet PLOS ONE’s publication criteria as it currently stands. Therefore, we invite you to submit a revised version of the manuscript that addresses the points raised during the review process.

Based on the comments from the reviewers and my own observation I recommend major revisions for this paper.

We look forward to receiving your revised manuscript.

Kind regards,

Thippa Reddy Gadekallu

Academic Editor

PLOS ONE

Journal Requirements:

3. Please include a copy of Table 3 which you refer to in your text on pages 5 and 6.  (NB. there are currently two tables labelled as Table 2.)

Reviewers' comments:

Reviewer's Responses to Questions

**Comments to the Author**

1. Is the manuscript technically sound, and do the data support the conclusions?

Reviewer #1: Yes

Reviewer #2: Yes

2. Has the statistical analysis been performed appropriately and rigorously? 

Reviewer #1: Yes

Reviewer #2: Yes

3. Have the authors made all data underlying the findings in their manuscript fully available?

Reviewer #1: Yes

Reviewer #2: Yes

4. Is the manuscript presented in an intelligible fashion and written in standard English?

Reviewer #1: No

Reviewer #2: Yes

5. Review Comments to the Author

Reviewer #1: The research presented by authors on the Evaluation of COVID-19 Predictions' Precision

with a Lyapunov-like exponent, the research is suitable for publication. However there are major revisions required in the paper.

1. Literature survey of the paper is too weak, the authors are advised to revise accordingly. Authors are advised to revise the methodology section and suggesting to provide comparative analysis to connect regarding the predictions precision/

2. English and Grammatical mistakes needs to be improved throughout the paper.

3. References needs to be in the format and authors are advised to refer the below articles and include the comparative analysis , build literature survey from the below papers

-> Gomathi, S., Kohli, R., Soni, M., Dhiman, G. and Nair, R. (2020), "Pattern analysis: predicting COVID-19 pandemic in India using AutoML", World Journal of Engineering, Vol. ahead-of-print No. ahead-of-print. https://doi.org/10.1108/WJE-09-2020-0450

-> K. Chandra, G. Kapoor, R. Kohli and A. Gupta, "Improving software quality using machine learning," 2016 International Conference on Innovation and Challenges in Cyber Security (ICICCS-INBUSH), Greater Noida, India, 2016, pp. 115-118, doi: 10.1109/ICICCS.2016.7542340.

Reviewer #2: 1. Abstract has to be re-written. The authors have to start with discussion on the importance of the topic, limitations of existing works and then introduce their solutions.

2. What are the limitations of the existing works that motivated the researchers to carry out the current work?

3. List out the major contributions of the current work.

4. SOme of the recent works on covid-19 and other disease predictions such as the following can be discussed in the paper "An Incentive Based Approach for COVID-19 using Blockchain Technology, Deep learning and medical image processing for coronavirus (COVID-19) pandemic: A survey, Early detection of diabetic retinopathy using PCA-firefly based deep learning model".

5. A detailed discussion on authors' inferences on the results obtained has to be discussed in the paper.

6. Discuss about the limitations of the existing works.

6. PLOS authors have the option to publish the peer review history of their article (what does this mean?). If published, this will include your full peer review and any attached files.

Reviewer #1: **Yes: **Rashi Kohli

Reviewer #2: No

---

## [Author Response · Author response to Decision Letter 0]

13 May 2021

Dear Editor and Reviewers,

I have attached responses to reviewers' comments in a separate file "Responses to Reviewers"

---

## [Decision Letter · Decision Letter 1]

17 May 2021

The Evaluation of COVID-19 Prediction Precision with a Lyapunov-like exponent

PONE-D-21-09821R1

Dear Dr. Mazurek,

We’re pleased to inform you that your manuscript has been judged scientifically suitable for publication and will be formally accepted for publication once it meets all outstanding technical requirements.

Kind regards,

Thippa Reddy Gadekallu

Academic Editor

PLOS ONE

Additional Editor Comments (optional):

Reviewers' comments:

Reviewer's Responses to Questions

**Comments to the Author**

1. If the authors have adequately addressed your comments raised in a previous round of review and you feel that this manuscript is now acceptable for publication, you may indicate that here to bypass the “Comments to the Author” section, enter your conflict of interest statement in the “Confidential to Editor” section, and submit your "Accept" recommendation.

Reviewer #1: All comments have been addressed

Reviewer #2: All comments have been addressed

2. Is the manuscript technically sound, and do the data support the conclusions?

Reviewer #1: Yes

Reviewer #2: Yes

3. Has the statistical analysis been performed appropriately and rigorously? 

Reviewer #1: Yes

Reviewer #2: Yes

4. Have the authors made all data underlying the findings in their manuscript fully available?

Reviewer #1: Yes

Reviewer #2: Yes

5. Is the manuscript presented in an intelligible fashion and written in standard English?

Reviewer #1: Yes

Reviewer #2: Yes

6. Review Comments to the Author

Reviewer #1: Paper is accepted. Authors did revisions based on the reviews. Paper is ready for publication . Authors have organized the paper for clarity.

Reviewer #2: I accept the paper for publication in its present form as the authors addressed all my comments in the paper.

7. PLOS authors have the option to publish the peer review history of their article (what does this mean?). If published, this will include your full peer review and any attached files.

Reviewer #1: No

Reviewer #2: No

---

## [Editor Report · Acceptance letter]

19 May 2021

PONE-D-21-09821R1 

The Evaluation of COVID-19 Prediction Precision with a Lyapunov-like exponent 

Dear Dr. Mazurek:

I'm pleased to inform you that your manuscript has been deemed suitable for publication in PLOS ONE. Congratulations! Your manuscript is now with our production department. 

Kind regards, 

on behalf of

Dr. Thippa Reddy Gadekallu 

Academic Editor

PLOS ONE